# iPLA2β-Null Mice Show HCC Protection by an Induction of Cell-Cycle Arrest after Diethylnitrosamine Treatment

**DOI:** 10.3390/ijms232213760

**Published:** 2022-11-09

**Authors:** Adriana Andrade, Tanja Poth, Alexander Brobeil, Uta Merle, Walee Chamulitrat

**Affiliations:** 1Department of Internal Medicine IV (Gastroenterology and Infectious Disease), University Hospital Heidelberg, 69120 Heidelberg, Germany; 2Center for Model System and Comparative Pathology, University Hospital Heidelberg, 69120 Heidelberg, Germany; 3Tissuebank of the NCT, Institute of Pathology, University Hospital Heidelberg, 69120 Heidelberg, Germany

**Keywords:** PLA2G6, liver cancer, nitrosamine, cell-cycle, knockout mice

## Abstract

Group VIA phospholipase A2 (iPLA2β) play diverse biological functions in epithelial cells and macrophages. Global deletion in iPLA2β-null (KO) mice leads to protection against hepatic steatosis in non-alcoholic fatty liver disease, in part, due to the replenishment of the loss of hepatocellular phospholipids. As the loss of phospholipids also occurs in hepatocellular carcinoma (HCC), we hypothesized that global deletion in KO mice may lead to protection against HCC. Here, HCC induced by diethylnitrosamine (DEN) was chosen because DEN causes direct injury to the hepatocytes. Male wild-type (WT) and KO mice at 3–5 weeks of age (12–13 mice/group) were subjected to a single intraperitoneal treatment with 10 mg/kg DEN, and mice were killed 12 months later. Analyses of histology, plasma cytokines, and gene expression were performed. Due to the low-dose DEN used, we observed a liver nodule in 3 of 13 WT and 2 of 12 KO mice. Only one DEN-treated WT mouse was confirmed to have HCC. DEN-treated KO mice did not show any HCC but showed suppressed hepatic expression of cell-cycle cyclinD2 and BCL2 as well as inflammatory markers IL-1β, IL-10, and VCAM-1. Notably, DEN-treated KO mice showed increased hepatic necrosis and elevated levels of plasma lactate dehydrogenase suggesting an exacerbation of liver injury. Thus, global iPLA2β deficiency in DEN-treated mice rendered HCC protection by an induction of cell-cycle arrest. Our results suggest the role of iPLA2β inhibition in HCC treatment.

## 1. Introduction

Hepatocellular carcinoma (HCC) is the second and sixth cause of cancer-related deaths in men and women, respectively [1]. In spite of the remarkable capacity of the liver to regenerate, most chronic liver injuries induce fibrosis, which can develop into cirrhosis and HCC. While inflammation is one of the key pathogenic mechanisms of HCC [2], metabolic alterations of phospholipids (PLs) seen in blood [3,4] and livers [5,6,7] of HCC patients are considered important events associated with HCC pathogenesis. PLs constitute the bulk of the membrane’s lipid matrix. Their synthesis and metabolism are highly controlled to maintain proper hepatocellular functions and integrity [8,9]. A decrease in PL contents can lead to chronic liver disease such as HCC and non-alcoholic fatty liver disease (NAFLD) in mice [10]. Consistently, a decrease in total PLs is observed in human HCC, predominantly, in males [6]. Human HCC shows an increase in PLs containing saturated fatty acids but specifically a decrease in those containing polyunsaturated fatty acids (PUFAs) [7]. Furthermore, a decrease in lysoPC is reported in human HCC [11] suggesting an important role of PL deacylation and reacylation in this disease. Taken together, hepatic PL synthesis and metabolism play a critical role in the development of HCC.

Group VIA calcium-independent phospholipase A2 (iPLA2β, PLA2G6 or PNPLA9) is a widely expressed enzyme that hydrolyzes PLs at the sn-2 position producing a free fatty acid and a lysoPL [12]. The hydrolysis allows reacylation of PUFAs into PLs that function to maintain membrane homeostasis. Global deletion in iPLA2β-null (KO) mice disturbs membrane homeostasis resulting in major phenotypes of male infertility and neurological disorders [12]. iPLA2 is shown to regulate cell growth [13]; however, the role of iPLA2β on HCC development is complex. In woodchuck hepatitis virus/c-myc mice [14], it is shown that suppressed iPLA2β contributes to chronic inflammation at an early stage; however, iPLA2β is enhanced at the late stage in supporting HCC development. The latter is inconsistent with the findings that global deletion in KO mice leads to protection against xenograft ovarian epithelial cancer [15] and development of mesenteric lymph node lymphoma [16]. On the contrary, we found that KO mice, which were housed in animal rooms infected with *novo* virus, *Helicobactor* spp., *Helicobactor typhlonius*, and *Helicobactor Hepaticus*, showed an increase in HCC incidence (unpublished results). We surmise that such infection-sensitized HCC could be due to hyperactivated macrophages by global iPLA2β deficiency previously observed in non-alcoholic steatohepatitis [17]. As the loss of hepatocellular PLs is seen in HCC [6,7,11], we hypothesize that these KO mice may be protected from HCC that is primarily induced by a direct injury to the hepatocytes. Here, the commonly used carcinogen, diethylnitrosamine (DEN), was chosen for HCC induction because it is metabolized to generate intermediates that react with nucleic acids resulting in genetic alterations and HCC formation [18]. We demonstrated that DEN-treated KO mice showed an increase in hepatic necrosis without any HCC detected. Thus, pan-iPLA2β inhibitors may be effective for the treatment of not only epithelial, such as ovarian cancers [15], but also HCC.

## 2. Results and Discussion

### 2.1. iPLA2β-Null Mice Show Reduced Body-Weight Gains, No HCC Detectable, and Enhanced Hepatic Necrosis after DEN Treatment

It is known that hepatic pathogenesis induced by DEN is dependent on the sex, age, and the strain of mice, and mice of male gender are recommended for DEN-mediated HCC [19,20]. Compared to other mouse strains, C57BL/6 mice are known to be resistant against HCC [20]; however, in our hands, a single intraperitoneal administration of five three-week-old male C57BL/6 mice with 50 mg/kg DEN was found to be toxic causing death one week later. In our study cohort, we therefore decided to use a much lower DEN dose at 10 mg/kg for treatment of male WT and KO mice at 3–5 weeks of age. Treated mice were harvested for analysis 12 months later. In our previous studies, male KO mice at 12 months old did not show any hepatic abnormalities [21]; therefore, we did not include WT and KO groups without DEN treatment in our cohort.

Due to the very low DEN dose used in our study, only a small size nodule was found in each liver of three from 13 DEN-treated WT (0.24, 0.34, and 0.42 cm diameter) and of two from 12 DEN-treated KO (0.16 and 0.27 cm diameter) mice (Figure 1a). Among these mice, only one mouse, i.e., DEN-treated WT#8 with a 0.34 cm nodule, showed characteristics of HCC cells (Figure 2a). In addition to the very low DEN dose used, mice at 3–5 weeks of age used in our cohort were older than the commonly used, 2 weeks of age [19,20]. We were not able to obtain mice at 2 weeks of age due to a delayed delivery of tail biopsies for genotyping. While 50 mg/kg DEN was toxic to our mice, it was difficult to judge whether 10 mg/kg DEN was insufficient to induce aggressive HCC observed 12 months later. Our animal license also did not include DEN dose-response studies. We therefore decided to perform phenotyping of mice under 10 mg/kg DEN treatment. After sacrifice, only a slight increase in heart weights was found in DEN-treated KO mice (Figure 1b). However, they showed lesser body weight gains in grams (Figure 1c) and % increase from starting body weights (Figure 1d) obtained from total number of mice (left panel) and mice without nodules (right panel). As mice exhibiting HCC are reported to show an increase in body weight gains [22], this parameter was consistently the highest in DEN-treated WT Mouse #8 (the circled value in Figure 1c,d). The observed reduction of this parameter in DEN-treated KO mice, particularly those with nodules, may be indicative of HCC reduction. Despite a few nodules detected in WT and KO mice (Figure 1a), but we were still able to study and compare the abnormalities in their livers by histology, plasma cytokines, and gene expression.

Upon histopathological evaluation of H&E-stained livers of DEN-treated WT mice [23,24], HCC at a grade 3 severity level was observed in the nodule of DEN-treated WT Mouse #8 (Figure 2a). A compression at the border line between HCC and normal tissue (indicated as NT) was observed together with massive lymphoplasmacellular infiltration. At high magnification shown in Figure 2a, macrovesicular lipid droplets (LD indicated by an arrow), cytoplasmic inclusions (CI indicated by dotted arrows), and lymphoplasmacellar infiltration could be observed among morphologically aberrant HCC cells. The nodules of two other DEN-treated WT (Mouse #5 and #13) displayed hyperplastic nodules without any compression characteristics of hepatocellular adenoma or HCC. While the nodule of WT Mouse #5 showed no hepatic abnormalities, that of WT #13 showed some lymphoplasmacellular infiltration (not shown).

Microvesicular steatosis was evident in DEN-treated WT mice with six mice showing grade 1 and four mice showing grade 2 level of steatosis. Grade 2 steatosis with macrovesicular LD was shown for WT Mouse #2 (Figure 2a). With one exception, WT Mouse #10 showed multiple large foci of heteromorphy lymphoid infiltrates beneath some central veins with a suspicion of lymphoma (Figure 2a). Hepatic necrosis at grade 1 and 2 was also observed in two DEN-treated WT mice. No significant abnormalities were detected in the duodenum and spleen of DEN-treated WT mice, while mild lymphoplasmacellular infiltration could be observed in the pancreas of DEN-treated WT with nodules, Mouse #5 and #8 (not shown).

None of DEN-treated KO mice with nodules displayed any compression of adjacent hepatic parenchyma characteristics of HCC or hepatocellular adenoma. Hyperplastic nodules were found in KO Mouse #7 (Figure 2b) and Mouse #10 (Figure 1a). The nodule of KO Mouse #7 displayed a subcapsular area with widespread hepatic necrosis (indicated as N), concomitant with chronic fibrosing active inflammation and microvesicular steatosis grade 1. In addition to KO Mouse #7, seven other DEN-treated KO mice showed hepatic necrosis with grade 1 (shown for KO Mouse #12) and grade 2 (shown for KO Mouse #5) (Figure 2b). By plotting necrosis scores of livers outside the nodules [23], DEN-treated KO mice without nodules showed a significant increase in hepatic necrosis (Figure 2c) and the elevation of plasma LDH (Figure 2d) when compared to DEN-treated WT mice without nodules. Thus, the deficiency may exacerbate DEN-induced liver injury independent of nodule formation. Among mice with nodules, no apparent difference in hepatic necrosis was observed between the genotypes. This indicates no correlation between hepatic necrosis and the formation of DEN-induced nodules observed in WT and KO mice. Overall microvesicular steatosis scores [24] were not different among DEN-treated WT and KO mice (not shown). No abnormalities were found in the pancreas, spleen, and duodenum of DEN-treated KO mice.

Thus, global iPLA2β deficiency led to a reduction of body-weight gains after DEN treatment together with no HCC phenotypes in the nodules. The deficiency however increased hepatic necrosis in mice that did not exhibit nodules. Thus, hepatic necrosis was not correlated with nodular development, but rather related to exacerbation of DEN-induced liver injury in mutant mice without nodules. In addition to reduced body weights, further analyses were needed to phenotype HCC protection in DEN-treated KO mice.

### 2.2. DEN-Treated iPLA2β-Null Mice Show Attenuated Cell-Cycle and Inflammatory Markers

Even though the majority of DEN-treated WT and KO (10 mice per genotype) did not exhibit any nodules, we continued to perform the analyses of mouse plasma and livers in order to understand KO phenotypes. The analyses of plasma inflammatory cytokines revealed an attenuation of IL-1β levels in DEN-treated KO mice obtained from total number of mice (left panel) and mice without nodules (right panel), while TNF-α, IL-6, IL-10, and IFN-γ levels were not affected (Figure 3a).

Upon analyses of inflammatory genes, DEN-treated KO mice consistently showed attenuated mRNA expression of vascular cell adhesion molecule-1 (*VCAM1*) obtained from the total number of mice and mice without nodules (Figure 3b). The deficiency attenuated the expression of *IL-1β* obtained from the total number of mice and *IL-10* from the total number of mice and mice without nodules (Figure 3c). IL-1β and IL-10 are recognized as pro-inflammatory mediators of HCC [25], and attenuated IL-10 may reflect reduced hepatocyte regenerative response [26]. Attenuated VCAM-1 expression may lead to the reduction of lymphocyte adhesion to liver endothelium [27] and an increase in apoptosis of inflammatory T cells [28]. Thus, the attenuated levels of *IL-1β*, *IL-10*, and *VCAM1* were consistent with the lack of HCC observed in DEN-treated KO mice (Figure 1 and Figure 2a,b).

iPLA2β deficiency, however, did not alter mRNA expression of apoptosis markers including apoptosis inducing factor (AIF), caspase3 (Casp3), high-mobility group box1 (HMGB1), BAX, BAD, and TLR3 (Figure 3d) as well as caspase 3/7 activities in liver homogenates (not shown). Remarkably, a significant decrease in mRNA expression of cyclin D2 (*CCND2*) and B cell lymphoma2 (*BCL2*) was observed in DEN-treated KO mice obtained from the total number of mice and mice without nodules (Figure 3d). BCL2 is a prognostic indicator in HCC for cell proliferation [29] and epithelial-mesenchymal transition [30]. Suppressed BCL2 is shown to decrease HCC survival by increasing non-apoptotic cell death including ferroptosis [31]. It is thus speculated that such non-apoptotic cell death may be one of the mechanisms for hepatic necrosis and elevation of plasma LDH seen in DEN-treated KO mice without nodules (Figure 2c,d). In line with this notion, genetic or pharmacological inactivation of iPLA2β has been shown to sensitize cells to ferroptosis [32]. As ferroptosis is a form of cell death occurring during therapy of HCC [33], it is thus warranted to further investigate ferroptosis as one of the mechanisms for protection against DEN-induced HCC by hepatocyte-specific iPLA2β deficiency.

CCND2 is one of the G1 cyclins that plays an important role in G1 cell-cycle transition, and their overexpression is implicated in neoplastic transformation [34]. A decrease in hepatic CCND2 expression in DEN-treated KO mice (Figure 3d) implies the presence of a cell-cycle arrest leading to suppressed proliferation of HCC cells. Indeed, suppressed mitosis/proliferation leads to the inability of hepatocytes to regenerate during chemically induced hepatotoxicity [35]. Consistent with the role of iPLA2 on cell growth [13], an inhibition of iPLA2β is shown to induce a cell-cycle arrest in the G1 phase [36], which involves an increase in PLs containing PUFAs [37]. Thus, HCC protection by iPLA2β deficiency could be, in part, due to a cell-cycle arrest through the replenishment of hepatic loss of PUFA-containing PLs.

In Figure 4, we attempted to outline the mechanisms for the reduction of HCC observed in DEN-treated mutants. DEN undergoes metabolism in the liver to produce diazonium ions as alkylating intermediates that react with nucleic acids and proteins to form alkylated macromolecules resulting in a sequence of genetic alterations that transform hepatocytes to HCC cells [18]. By the generation of acetaldehyde and oxidative stress, DEN also initiates hepatocyte injury, inflammation, and necrosis. In response to DEN-induced injury, hepatocytes undergo mitosis for proliferative regeneration, and some selected hepatocytes with genetic alterations may become precursor cells [35,38], including tumor-initiating LGR5 stem cells [39] to form nodules and subsequently HCC.

It is known that PLA2 enzymes participate in mobilization of PUFAs [40] (Figure 4). Cytosolic PLA2 (*cPLA2*) has specificity to cleave arachidonic acid (AA), which is a substrate of cyclooxygenase 2 (COX2) to generate inflammatory eicosanoids and prostanoids, such as prostaglandin E2 (PGE2). It has been reported that iPLA2β does not couple with COX2 [41]; this decoupling is consistent with the attenuation of hepatic PGE2 in *ob*/*ob* mice by iPLA2β deficiency [21]. Similarly, hepatic PGE2 may be attenuated in DEN-treated KO mice, and this attenuation may limit the expansion of LGR5 stem cells leading to HCC protection [42]. iPLA2β preferentially mobilizes ω-3 PUFAs such as docosahexaenoic acid (DHA) [40], which is shown to inhibit HCC by blocking β-catenin and COX2 [43]. DHA is also a precursor of pro-resolution lipids, Resolvins, which are reported to reduce the stemness of HCC [44]. Consistently, ResolvinD2 is shown to be elevated in peritoneal macrophages of KO mice [45]. Furthermore, we recently discovered that hepatocyte-specific iPLA2β deletion led to an increase in hepatic Lipoxin A4 [46], which is shown to suppress HCC via remodeling of tumor microenvironment including LGR5+ compartments [47]. Thus, the proposed elevation of these pro-resolution lipids in DEN-treated KO mice may lead to attenuated inflammation leading to reduced nodular formation and hence HCC protection (Figure 4).

At the cell membrane, iPLA2β deficiency in hepatocytes leads to an accumulation of PLs containing PUFAs [6,7,11], and this would prevent the loss of PLs occurring during preneoplastic progression to HCC (Figure 4). Moreover, these PLs may induce a cell-cycle arrest [36,37], leading to a decrease in regenerative response [35,38]. Alternatively, these membrane PUFA-PLs can be oxidized by 5/15-lipoxygenase to generate corresponding hydroperoxides [PLs(PUFAOOH)], of which in the presence of cellular iron have been identified as mediators of ferroptosis [32]. Increased hepatic ferroptosis/necrosis in DEN-treated KO mice may hamper mitosis and proliferation of preneoplastic cells [35,38].

Regarding the effects of iPLA2β deficiency in macrophages on hepatic inflammation, current data have shown that macrophages or Kupffer cells from KO mice displayed suppressed M1 cytokines, such as, TNF-α and IL-6 [16]. However, Kupffer cells from KO mice with Jo-2-induced liver injury in turn released elevated levels of IL-6 indicating opposing changes towards M1 during injury. For further understanding, mice with macrophage-specific iPLA2β deletion have been generated in our laboratory and initial report has been published in an abstract form [48]. Male macrophage-specific mutants at 12 months of age displayed hepatic necrosis, elevated plasma IL-6, and interstitial infiltration of immune cells in vivo. These results indicate the propensity of iPLA2β-deficient macrophages towards M1 activation perhaps via an induction of ferroptosis in these cells [32]. More relevantly, the promotion of M1-macrophage response is shown to induce protective tumor immunity [49] and works in concert with suicide gene therapy and cisplatin to improve the treatment of HCC [50] and lung cancer [51], respectively. Thus, HCC protection observed in DEN-treated KO mice could be due to the response of the gene deletion specifically in macrophages. Taken together, iPLA2β inactivation in hepatocytes via cell-cycle arrest [36,37] and macrophages via M1 [16,48] in DEN-treated KO mice may contribute to hepatic necrosis as a background. Upon DEN treatment, this background could lead to suppression of mitosis and HCC regeneration resulting HCC protection. In opposition to iPLA2β deficiency in macrophages [16,48], the deficiency in hepatocytes may render protection against hepatic inflammation leading to attenuated preneoplastic regeneration and HCC protection (Figure 4).

Concurrently, unspecific inhibitors of calcium-independent PLA2 have widely been tested in vitro in epithelial breast [52], prostate [53], and intestinal [54] cancer cells. iPLA2β-specific inhibitor, FKGK11, is able to inhibit epithelial ovarian cancer development in xenograft models [55]. Multiple administration of the iPLA2 inhibitor, bromoenol lactone, combined with an anti-cancer drug, paclitaxel, also reduces the number of ovarian tumors [55]. Hence, pan-iPLA2β inhibitors may be therapeutically useful for epithelial cancers including HCC. We speculate that these pan-inhibitors may be less effective, when compared with non-viral HCC, to treat infection/viral-induced HCC due to overt macrophage hyperactivation [16,17,38]. Further experiments are necessary to determine whether FKGK11 as well as hepatocyte- or macrophage-specific deletion of iPLA2β could modulate HCC induced by DEN or DEN combined with high-fat-diet feeding. These results could provide some clues whether pan-iPLA2β inhibitors or hepatocyte-, macrophage-specific iPLA2β inhibitors can represent a class of drugs for therapy of non-viral HCC.

## 3. Materials and Methods

### 3.1. Animals and Treatment

KO mice were gifts from Dr. John Turk (Washington University School of Medicine, St. Louis, MO, USA). KO mice were bred and genotyped according to published methods [16,17,18,19,20]. KO mice were bred into C57BL/6 background for at least 20 generations. All mice were bred and kept in the Interfaculty Biomedical Facility of the University of Heidelberg (Im Neuenheimer Feld 347). Male KO (N = 12) and control C57BL/6 (WT, N = 13) mice at 3–5 weeks old were intraperitoneally injected with 10 mg/kg DEN and were killed 12 months later. Body and organ weights were recorded before and after DEN treatment. Blood was collected and liver, spleen, and intestine were fixed in formalin and snap frozen in liquid nitrogen. Studies involving animals have been approved by the University of Heidelberg Institutional Animal Care and Use Committee. This study protocol was reviewed and approved by German Authority in Karlsruhe (Baden-Württemberg Regierungspräsidium Karlsruhe) with approval number G248/11, according to Animal Welfare Laboratory Animal Ordinance (Tierschutz-Versuchstierverordnung, TierSchVersV) from the German Animal Welfare Act (Tierschutzgesetz, TierScG).

### 3.2. Biochemical Assays

Plasma lactate dehydrogenase (LDH) activities were measured using Randox kits (Krefeld, Germany). Liver homogenates were prepared for determination of caspase 3/7 activity using caspase 3/7^Glo^ kit (Promega, Mannheim, Germany) with a luminometer Lumat LB 9507 (Berthold Technologies, Bad Wildbad, Germany). Caspase 3/7 activity was normalized to mg protein.

### 3.3. Histology

After overnight fixation in 10% buffered formalin representative specimens of liver, spleen, pancreas, and duodenum were routinely dehydrated, embedded in paraffin, and then cut into 4-μm thick sections. The tissue sections were stained with H&E according to standard protocols. Hepatocellular responses including fatty changes, hepatocellular necrosis, and inflammatory cell infiltration were identified according to published procedures [23,24]. The grading for severity of hepatic necrosis [23] was performed by Drs. Poth and Brobeil using the following scoring system: grade 1 (1–2 foci as minimal), grade 2 (3–6 foci as few), grade 3 (7–12 foci as several), grade 4 (>12 foci as many), and grade 5 (diffuse, severe). Steatosis was scored according to a published protocol by Kleiner et al. [24].

### 3.4. ELISA

Plasma samples were subjected to determination of cytokines TNF-α, IL-6, IL-10, INF-γ, and IL-1β using ELISA kits from Biolegends (Cologne, Germany).

### 3.5. Gene Expression

RNA was isolated from liver samples using Gen Elute Miniprep kit from Sigma (Taufkirchen, Germany). cDNAs were prepared from RNA using a Thermo Scientific’s cDNA kit (Karlsruhe, Germany). mRNA expression was analyzed on an Applied Biosystems 7500 using TaqMan^®^ assay-on-demand primers. The expression level was calculated using Δ−Ct transformation method and determined as a ratio of target gene with house-keeping gene GAPDH.

### 3.6. Statistics

Data was presented as mean ± SEM. By using GraphPad Prism 7 (GraphPad, La Jolla, CA, USA), statistical analysis was performed using the Mann-Whitney U tests with *p* < 0.05 considered significant or *t*-tests (*p* = 0.08) with Welch’s correction with *p* < 0.1 considered significant (Figure 3).

## 4. Conclusions

In conclusion, global iPLA2β deficiency in DEN-treated mice limited HCC development with mechanisms involving the induction of cell-cycle arrest and attenuation of inflammatory markers Bcl2, IL-1β, IL-10, and VCAM-1. These results are in line with the reported effects of iPLA2β inactivation in hepatocytes and macrophages. Thus, pan- iPLA2β inhibitors may be effective for treatment of not only epithelial ovarian cancers [15,46] but also HCC.

## Figures and Tables

**Figure 1 ijms-23-13760-f001:**
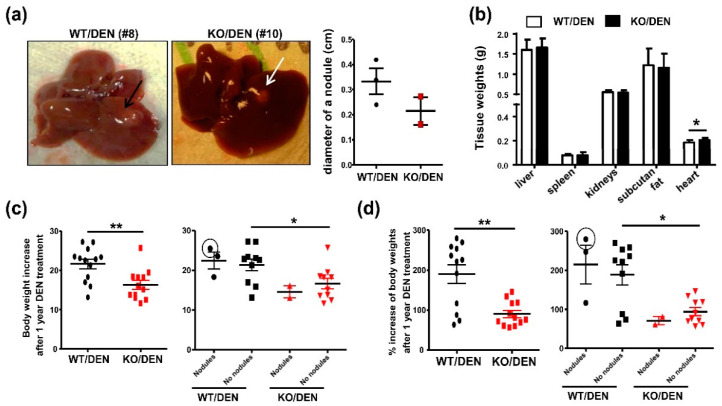
Mice at 3–5 weeks old were treated with a single dose of 10 mg/kg DEN and sacrificed 12 months later. Tissue weights and the diameter of any observable nodules were measured. (**a**) Left panel shows pictures of a small nodule of DEN-treated WT (Mouse #8, indicated by a black arrow) and KO (Mouse #10, indicated by a white arrow). Right panel shows a plot of nodule diameters of 3 DEN-treated WT mice and 2 of DEN-treated KO mice. (**b**) Absolute liver, spleen, kidneys, subcutaneous fat, and heart weights. (**c**) An increase in body weight in grams after one-year DEN treatment obtained from total number of mice (left) and mice with ‘Nodules’ or ‘No nodules’ (right). (**d**) % increase from starting body weights obtained from total number of mice (left) and mice with ‘Nodules’ or ‘No nodules’ (right). The value with a black circle was from DEN-treated WT Mouse #8. Data are mean ± SEM, N = 12–13 for (**b**–**d**). * *p* < 0.05, ** *p* < 0.01 with Mann-Whitney U tests.

**Figure 2 ijms-23-13760-f002:**
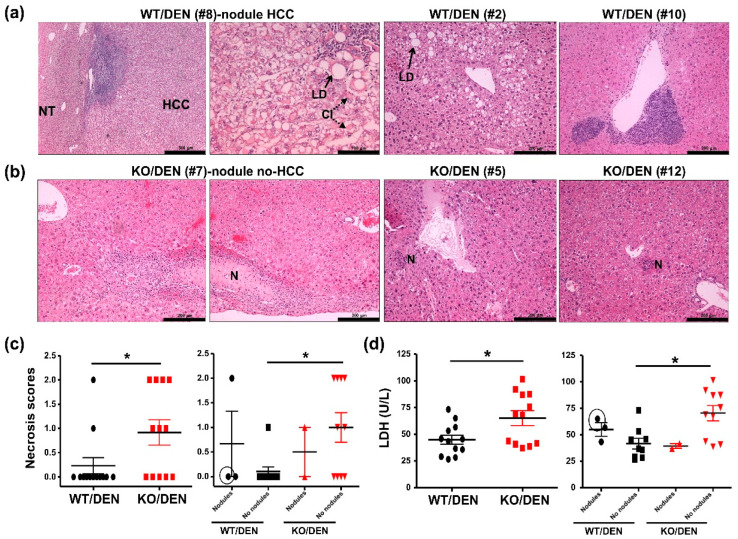
Mice were treated as described in Figure 1. (**a**) Histological evaluation of WT Mouse #8 mice treated with DEN showed tumor (HCC) bordering to normal tissue (NT). HCC cells were mixed with lipid droplets (LD indicated by a solid arrow), cytoplasmic inclusions (CI indicated by dotted arrows), and lymphoplasmacellar infiltration. Other abnormalities in DEN-treated WT mice were fatty changes (Mouse #2) and heteromorphous population of lymphocytes and heteromorphous population of lymphocytes suspicious for lymphoma (Mouse #10). (**b**) Histological evaluation of livers of DEN-treated KO mice showed a compressed necrotic (N) area in hyperplastic nodule of Mouse #7. Focal hepatic necrosis was evident in Mouse #5 and Mouse #12. (**c**) Necrosis histology scores in livers outside the nodules obtained from total number of mice (left) and mice with ‘Nodules’ or ‘No nodules’ (right). (**d**) Plasma LDH (U/L) levels obtained from total number of mice (left) and mice with ‘Nodules’ or ‘No nodules’ (right). The value with a black circle was from DEN-treated WT Mouse #8. Data are mean ± SEM; N = 12–13. *, *p* < 0.05 with Mann-Whitney U tests.

**Figure 3 ijms-23-13760-f003:**
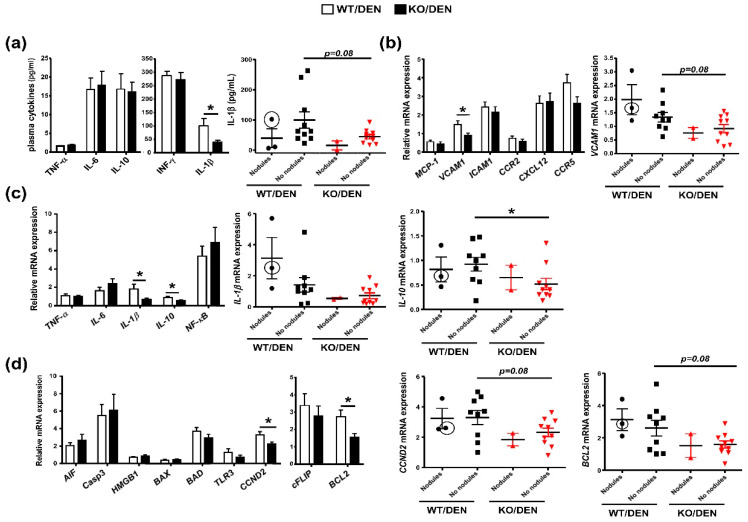
Mice were treated as described in Figure 1. Data in the left-hand panel were obtained from the total number of mice, and those in right-hand panel(s) were obtained from ‘Nodule’ and ‘No-nodule’ groups. (**a**) Plasma levels (pg/mL) of cytokines TNF-α, IL-6, IL-10, IFN-γ, and IL-1β. (**b**) Hepatic mRNA expression of genes related to chemokines (*MCP-1*, *VCAM1*, *ICAM1*, *CCR2*, *CXCL12*, and *CCR5*). (**c**) Hepatic mRNA expression of inflammatory genes (*TNF-α*, *IL-6*, *IL-1β*, *IL-10*, and *NF-kB*). (**d**) Hepatic mRNA expression of genes related to apoptosis (inducing factor (*AIF*), caspase3 [*Casp3*], high-mobility group box1 [*HMGB1*], *BAX*, *BAD*, and *TLR3*) as well as proliferation and anti-apoptosis (cyclin D2 [*CCND2*], *cFLIP*, and *BCL2*). The value with a black circle was from DEN-treated WT Mouse #8. Data are mean ± SEM; N = 12–13. * *p* < 0.05 with Mann-Whitney U tests for total number of mice, and *t*-tests (*p* = 0.08) with Welch’s correction for mice with or without nodules (*p* < 0.1 considered significant).

**Figure 4 ijms-23-13760-f004:**
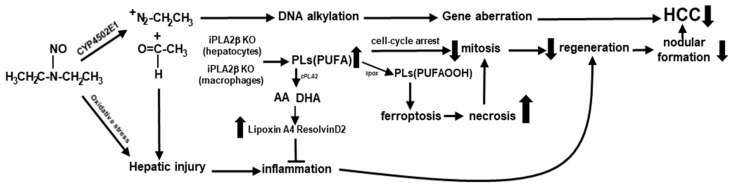
iPLA2β deficiency in hepatocytes results in an increase in hepatic PUFA-containing PLs leading to an induction of a cell-cycle arrest, suppression of HCC regenerative response, and induction of hepatic necrosis and ferroptosis likely via lipoxygenase-induced oxidation of PLs [PLs(PUFAOOH)]. Similarly, iPLA2β deficiency in macrophages may lead to ferroptosis. As a background, ferroptosis may lead to hepatic necrosis which may limit mitosis and liver cell regeneration. PLs(PUFA) are also precursors of pro-resolution lipoxin A4 and resolvinD2 leading to an attenuated inflammatory response resulting in reduced liver cell regeneration, reduced nodular formation, and HCC protection.

## Data Availability

The data that support the findings of this study are openly available online: https//www.wcmat.com/stats/ (accessed on 9 October 2022).

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
