# Peer review of "iPLA2β-Null Mice Show HCC Protection by an Induction of Cell-Cycle Arrest after Diethylnitrosamine Treatment"

_ijms, 2022, doi:10.3390/ijms232213760_

Round 1

Reviewer 1 Report (Previous Reviewer 1)

the manuscript is well-written and presents the research data in a scientifically detailed manner. the manuscript can be accepted after completion of all the codal formalities.

Author Response

Thank you for the review. We have edited the manuscript for style and double-checked the spelling.

Reviewer 2 Report (New Reviewer)

- Tumors in the model employed by the authors derive from the LGR5+ compartment (Cao et al,  LGR5 marks targetable tumor-initiating cells in mouse liver cancer. Nat Commun. 2020 Apr 23;11(1):1961. doi: 10.1038/s41467-020-15846-0), hence effects in this compartment are the relevant ones, also with respect to cell cycling. Can the authors say something on the role of this compartment in their results?

-The model used by the authors leads to oncogenic mutations in the armadillo repeats 5 and 6 of β-Catenin thar reduce binding to APC (Liu et al. Gastroenterology 2020 Mar;158(4):1029-1043). How does this relate to the results?

- Was histology scored by a qualified (murine) pathologist?

- In relation to the PUFA-related observations, effects of PLA2 likely derive from altered production of prostanoids and prostanoid signal transduction (Int J Biochem Cell Biol. 2004 Jul;36(7):1187-205), the potential links here should be discussed.

Author Response

Reviewer#2:

  1. In relation to the PUFA-related observations, effects of PLA2 likely derive from altered production of prostanoids and prostanoid signal transduction (Int J Biochem Cell Biol. 2004 Jul;36(7):1187-205), the potential links here should be discussed.

Our Response: Thank you for this helpful comment and we have modified Figure 4 accordingly. We have discussed more on the replenishment of PUFA-PLs by the deficiency related generation of PUFAs that can be precursors of pro-resolution lipis including Resolvin D2 and Lipoxin A4. These lipids may attenuate cancer stem cells and b-catenin pathways. These added yellow highlighted texts appear in lines 243-260.

  1. Tumors in the model employed by the authors derive from the LGR5+ compartment (Cao et al, LGR5 marks targetable tumor-initiating cells in mouse liver cancer. Nat Commun. 2020 Apr 23;11(1):1961. doi: 10.1038/s41467-020-15846-0), hence effects in this compartment are the relevant ones, also with respect to cell cycling. Can the authors say something on the role of this compartment in their results?

Our Response: Thank you for this comment. We have also discussed that the deficiency may attenuate hepatic PGE2 which is reported as a mediator of LGR5. The elevation of hepatic lipoxins in KO mice may suppress preneoplastic microenvironemtn including LGR5+ compartments. These texts are shown in lines 248-251 and line 258.

  1. The model used by the authors leads to oncogenic mutations in the armadillo repeats 5 and 6 of β-Catenin thar reduce binding to APC (Liu et al. Gastroenterology 2020 Mar;158(4):1029-1043). How does this relate to the results?

Our Response: Thank you for this comment. We have also discussed that the deficiency may attenuate hepatic β-Catenin by the action of metabolites from omega-3 PUFAs DHA as seen in line 253.

  1. Was histology scored by a qualified (murine) pathologist?

Our Response: Yes, the co-authors Dr. Poth and Dr. Brobeil evaluated all histology slides. We have added a statement relating to this in line 331.

This manuscript is a resubmission of an earlier submission. The following is a list of the peer review reports and author responses from that submission.

Round 1

Reviewer 1 Report

the manuscript describes the data on iPLA2 null mice and HCC severity after diethylnitrosamine treatment. the study is well described.

1. Introduction is a bit lengthy and can be more focused. 

2. there are few syntax errors, otherwise the manuscript is worth publication.

Author Response

Reviewer # 1

  1. Introduction is a bit lengthy and can be more focused.

Response: The Introduction has been shortened to focus on the role of hepatocyte iPLA2b in the development of HCC (rather than NAFLD/NASH). Some references realted to NAFLD/NASH have been removed.

  1. There are few syntax errors, otherwise the manuscript is worth publication

Response: Thank you. We have spoted many syntax errors in using the language. We have made corrections as shown in yellow highlighted texts in lines 13, 21, 24-26, 37-39, 51, 54, 66-67, 203, and 205-207.

Reviewer 2 Report

Dr. Adriana Andrade, et al try to address the importance of iPLA2beta for HCC induction.

While the roles of iPLA2beta which hydrolyzes PLs and regulates cell growth on HCC development are interesting, however, there are some limitations for publication.

 1.     The incidence of HCC is too low. The protocol they used is not appropriate. As they mentioned, two weeks old mice and 25mg/kg of DEN are commoly used. They should adopt these condition with permission for the use of DEN-HCC model.

2.     The significance of HCC severity they analyzed is not sufficient.

3.     They should check more molecules and marekers related to cell death and cell proliferation than ones shown in Fig3.

4.     Inconsinstent descriptions. In the introduction part, pan-iPLA2beta inhibitiors are promise for HCC therapy, but skeptical in the discussion part.

5.     They should use pan-iPLA2beta inhibitiors for the treatment of HCC in the mouse model.

6.     NASH or NAFLD models should be investigated. At least, a high-fat diet can be fed on the DEN-induced HCC mice.

Author Response

Reviewer # 2

  1. The incidence of HCC is too low. The protocol they used is not appropriate. As they mentioned, two weeks old mice and 25mg/kg of DEN are commoly used. They should adopt these condition with permission for the use of DEN-HCC model.

Response: Due to a delay in obtaining tail biopsies of new pubs from our animal facility, we were unable to genotype and identify iPLA2β-mutant mice quickly enough, therefore, we could not use them at two weeks of age. This is a concurrent logistic problem because we do not have direct control over the new pubs as we rely on workers at our animal facility. We added a sentence to explain this problem in lines 92-94.    

  1. The significance of HCC severity they analyzed is not sufficient.

Response: Admittedly, HCC severity was low because the age of mice and low-dose DEN used. HCC incidence induced was one in 13 WT mice and none in 12 mutant mice. With these 12-13 mice/group in our cohort, we were able to confidently demonstrate that the treated-mutant mice showed significant hepatic necrosis and elevated release of LDH. We have now emphasized the role of iPLA2b deficiency on hepatic necrosis as the primary response followed by the reduction of HCC (rather than HCC severity). The edited texts now appear in the title (lines 2-3) and lines 25, 67, 70, 134, 157-158, 162, 170, 217-218, 256, and 317-318. 

  1. They should check more molecules and marekers related to cell death and cell proliferation than ones shown in Fig3.

Response: In Fig.3, we analyzed 9 genes related to apoptosis, anti-apoptosis, and proliferation as well as capase 3/7 activity in liver homogenates. We were unable to observe any differences. This is due to the fact that liver tissues outside the nodule in treated-WT and mutant mice were normal tissues. This point is already indicated in lines 214-216.

  1. Inconsinstent descriptions. In the introduction part, pan-iPLA2beta inhibitors are promise for HCC therapy, but skeptical in the discussion part.

Response: Thank you for this comment. The skepticism was because that the deletion of iPLA2β specifically in macrophages may render an opposing outcome in exacerbating injury as we observed in other liver disease models (unpublished data). In this context, pan-iPLA2beta inhibitors may not be as effective to treat infection/viral-induced HCC compared to non-viral HCC. This text is now indicated in lines 263-265.

  1. They should use pan-iPLA2beta inhibitiors for the treatment of HCC in the mouse model.

Response: Thank you for this comment. To do this, we have to apply for an animal licence for administration of FKGK11 and DEN. This project will take 2-2.5 years. For discussion, we added a possibility to perform this experiment as shown in line 266.

  1. NASH or NAFLD models should be investigated. At least, a high-fat diet can be fed on the DEN-induced HCC mice.

Response: Thank you for this comment. The combination of high-fat-diet and DEN administration significantly has been shown to shorten the time for HCC development to 6 months. This option is now included in the discussion as shown in line 267.
